# Chemical Composition and In Vitro Antioxidant Activity and Anti-Acetylcholinesterase Activity of Essential Oils from *Tadehagi triquetrum* (L.) Ohashi

**DOI:** 10.3390/molecules28062734

**Published:** 2023-03-17

**Authors:** Wenzhi Song, Ziyue Xu, Peizhong Gao, Xu Liu

**Affiliations:** 1SDU-ANU Joint Science College, Shandong University, Weihai 264209, China; 2Marine College, Shandong University, Weihai 264209, China

**Keywords:** *Tadehagi triquetrum* (L.) ohashi, essential oils, chemical composition, antioxidant activity, anticholinesterase activity, GC-MS, GC-FID

## Abstract

The present study aimed to determine the chemical compositions of essential oils (EOs) from *Tadehagi triquetrum* (L.) Ohashi and evaluate their antioxidant and anti-cholinesterase activity under the comprehensive influence of chemical components. The essential oils were extracted from *T. triquetrum* (L.) Ohashi by hydrodistillation. A total of 58 organic compounds were identified by GC-FID and GC-MS analysis. The major components of *T. triquetrum* (L.) Ohashi EOs were identified as palmitic acid (22.46%), 1-Octen-3-ol (14.07%), Caryophyllene (7.20%), (Z)-18-Octadec-9-enolide (6.04%), and 3-Hexen-1-ol (4.55%). The antioxidant activity of the essential oils was determined by using ABTS assay, DPPH assay, and FRAP assay, with IC_50_ values of 2.12 ± 0.05 mg/mL, 4.73 ± 0.91 mg/mL against the ABTS, DPPH, and FRAP value 117.42 ± 8.10 mM/g. The result showed that it had moderate antioxidant activities in the experiment, which why it is likely that it will be used as an antioxidant. At the same time, the EOs also showed moderate anti-acetylcholinesterase activity. This study expands the chemical and biological knowledge of the EOs of *T. triquetrum*.

## 1. Introduction

Compounds that can inhibit the formation of free radicals or block their propagation by removing the species causing peroxidation, chelating metal ions, or destroying the autoxidative chain reaction are called antioxidants [1,2]. Antioxidants are widely used in the food industry and cosmetics industries. However, synthetic antioxidants may be toxic, since they may cause cancer and hurt human health [3]. In addition, global public health is facing severe challenges due to the excessive use of synthetic chemicals in today’s world, so people have turned more of their attention to natural substances [4]. Moreover, compared with synthetic oxidants, the natural antioxidant active substances in plant extracts may have the same or higher antioxidant effect as synthetic antioxidants, and natural antioxidants might be more harmless to the human body [3]. Compounds with antioxidant activity can be extracted from wild-growing or standardized plants [5]. Therefore, extracting antioxidant substances from plants may play an active role in replacing synthetic chemicals. People are increasingly interested in extracting natural antioxidants from plants [4]. Studying the antioxidant activity of essential oils from plant extracts is increasingly important [6].

The clinical significance of acetylcholinesterase lies in the fact that its content is the main indicator of liver function in the study. In human medical examination, the high content of acetylcholinesterase may reflect hyperthyroidism, chronic renal failure, diabetes, and other diseases. Therefore, the study of the anti-acetylcholinesterase activity of natural plants is of practical medical importance and opens up the possibility of finding safer drugs. Alzheimer’s disease (AD) is one of the diseases with the highest incidence rate among the elderly population [7]. Although the etiology of AD is unclear, reducing high levels of acetylcholine using acetylcholinesterase inhibitors has become the most effective treatment [8]. In addition, previous studies pointed out that protein A found in the aging spots of the AD brain can induce the release of oxygen-free radicals during inflammation, which can damage cell components [9,10]. Antioxidants can also attenuate the inflammatory pathway by scavenging oxygen-free radicals. The relationship between antioxidant activity and anti-acetylcholinesterase activity is worth exploring. 

*Tadehagi* (Fabaceae), which represented as shrub or subshrub, comprises about six species and is widespread in tropical and subtropical Asia, Pacific Islands, and Australia. There are two species in China, which are distributed in the provinces south of the Nan Mountains. The type of species of this genus, *T. triquetrum* (L.) H. Ohashi, growing primarily in the subtropical biome, bears bluish violet flowers and lanceolate leaves (Figure 1). *T. triquetrum* is a reputed medicinal plant, traditionally used against several diseases. As a traditional Chinese herbal medicine, *T. triquetrum* (L.) Ohashi has been widely used in folk medicine to lower fevers, expel toxic substances, promote urination, remove jaundice, prevent congestion, and kill insects. People often use the fish paste made with it to avoid the development of fly larvae in Myanmar [11]. The existing research shows the bioactive compounds in *T. triquetrum* (L.) Ohashi and its physiology and pharmacology are worthy of further study [11]. In previous studies, it has been shown that *T. triquetrum* (L.) Ohashi’s leaf extract contains various compounds that are effective ingredients to combat allergic asthma [12]. At the same time, through pathological examination, it was found that it also has an antihepatotoxic effect, inhibiting inflammation and liver cell damage to a certain extent [13]. Phenylpropanoid glucosides tadehaginosides C–J (73–80) and tadehaginoside (81) extracted from *T. triquetrum* (L.) Ohashi showed their potential as anti-diabetes compounds [14], which shows the profound role of *T. triquetrum* (L.) Ohashi in pathological research.

The medical efficacy of *T. triquetrum* (L.) Ohashi mentioned above is mainly related to oxidative stress, and *T. triquetrum* (L.) Ohashi is also a traditional Chinese medicine in folk medicine, showing an ethnomedicinal importance that cannot be ignored. It is worthwhile to identify its extract and evaluate its antioxidant activity. However, the existing literature still lacks the analysis of the chemical compositions of EOs and the biochemical activity (antioxidant activity, anticholinesterase activity, etc.) of essential oils extracts in *T. triquetrum* (L.) Ohashi. Therefore, to fill the research gap in this field and serve a broader range of scientific chemical research and analysis, this paper will focus on identifying the chemical composition of *T. triquetrum* (L.) Ohashi and evaluating the antioxidant activity and anti-acetylcholinesterase activity of its EOs extracts. Based on the existing application of the above, this experiment will complement the research on the biological activity behind its action.

## 2. Results

### 2.1. Identification of Chemical Compositions of Essential Oils

In this experiment, the essential oils of the stem section were extracted by a Clevenger-type apparatus. A total of 0.3 mL essential oil was obtained from 1000 g *T. triquetrum* (L.) Ohashi by hydrodistillation (yield 0.03%). The essential oils of *T. triquetrum* (L.) Ohashi is yellow, hydrophobic, and has a unique scent. Figure 2 shows the ion chromatogram of the essential oils of *T. triquetrum* (L.) Ohashi. 

GC-MS identified various classes of chemical components in EOs. According to the elution sequence on the chromatographic column, Table 1 lists the specific chemical components, retention time (RT), retention index (RI), percentage (%), and identification method in detail in the EOs of *T. triquetrum* (L.) Ohashi.

A total of 58 compounds were detected in the essential oils, including phenols, terpenoids, ketones, etc. The identified compounds accounted for 99.79% of the extracted essential oils. The proportion of each compound was accurately calculated by GC-FID, mainly including palmitic acid (22.70%), 1-Octen-3-ol (14.22%), Caryophyllene (7.27%), and (Z)-18-Octadec-9-enolide (6.10%). 

### 2.2. Antioxidant Activity of Essential Oils

Antioxidant activity involves a variety of reaction characteristics and mechanisms. The more common is free radical scavenging and reduction characteristics. ABTS (2,2′-Azinobis-(3-ethylbenzthiazoline-6-sulphonate)) and DPPH (1,1-Diphenyl-2-picrylhydrazyl radical 2,2-Diphenyl-1-(2,4,6-trinitrophenyl)hydrazyl) analysis can define the ability of the substance involved in the reaction to scavenge free radicals, while FRAP (ferric reducing ability of plasma) can detect the reducing ability of the antioxidant [15]. Therefore, different research methods can be used to comprehensively understand the antioxidant activity of the *T. triquetrum* (L.) Ohashi essential oil. ABTS, DPPH, and FRAP were used in this study to assess the antioxidant activities.

#### 2.2.1. ABTS Assay

ABTS is an experimental method widely used in the biological testing of antioxidant activity [16]. The results showed that the ABTS· free radical scavenging capacity of the essential oils of *T. triquetrum* (L.) Ohashi expressed significant dose dependence. The shielding effect of the EOs can represent the dose dependence in the ABTS free radical scavenging experiment in Figure 3. The ABTS IC_50_ was 2.12 ± 0.05 mg/mL. The IC_50_ of standard antioxidant BHT and Trolox was 14.69 ± 0.27 μg/mL and 9.47 ± 0.52 μg/mL, respectively, according to the same experimental procedure.

#### 2.2.2. DPPH Assay

This experiment adopted a simple procedure, quick results, and a complete evaluation method to show essential oils’ free radical scavenging activity [17]. The ability to scavenge free radicals was measured by DPPH. Figure 4 shows a dose-dependent relationship between the DPPH· free radical scavenging rate (RSA%) and the added concentration of *T. triquetrum* (L.) Ohashi essential oil. The increasing concentration of essential oils increased its free radical scavenging rate. According to the calculation, the IC_50_ value of essential oil was 4.73 ± 0.91 mg/mL. Under the same conditions, the IC_50_ values of the standard antioxidant BHT and Trolox were 42.90 ± 1.20 μg/mL and 11.82 ± 0.83 μg/mL, respectively. 

#### 2.2.3. FRAP Assay

The ferric reducing/antioxidant power (FRAP) assay was first used in 1996 to test the reducing ability of substances [18]. This method can be realized on the basis that the participating antioxidant in the test can reduce Fe (III) to Fe (II) [15]. It can be easily used to evaluate the antioxidant capacity of plants. In this experiment, an FRAP assay was used to identify the antioxidant power of the essential oils of *T. triquetrum* (L.) Ohashi as 117.42 ± 8.10 mM/g. The antioxidant activity of *T. triquetrum* (L.) Ohashi detected in this experiment is of particular practical significance.

### 2.3. Antiacetylcholinesterase Activity

The effective way to treat Alzheimer’s disease (AD) is to increase the content of the acetylcholinesterase inhibitor or acetylcholine [19]. In this experiment, the anti-acetylcholinesterase activity of the *T. triquetrum* (L.) Ohashi EOs was evaluated by detecting its inhibition of acetylcholinesterase (AChE) enzymes. In the AChE experiment, the inhibition of *T. triquetrum* essential oils expressed at the highest test concentration was 12.2%, showing moderate anti-acetylcholinesterase activity. Its IC_50_ value was greater than 250 μg/mL, as shown in Table 2.

## 3. Discussion

In this study of the essential oils of *T. triquetrum* (L.) Ohashi, the classification of various compositions deserves attention, as shown in Figure 5. Among the compositions, alcohols (including alcohols and enols) (34.66%) expressed the highest content, followed by fatty acids (25.09%), olefins (14.20%), and esters (10.82%).

Palmitic acid is the most important component in essential oils, so *T. triquetrum* (L.) Ohashi showed palmitic acid chemotype characteristics, which has far-reaching biological and chemical practical significance. In previous studies, palmitic acid extracted from marine red algae had selective cytotoxicity against human leukemia cells. Still, it was shown to be non-toxic to normal human dermal fibroblasts (HDF) cells in vitro. In further research, palmitic acid was detected to have a significant effect on DNA topoisomerase I without affecting DNA topoisomerase II, which makes palmitic acid a good prospect in the production of anti-cancer drugs [20]. At the same time, the study on the tumor inoculated with B16-F10 cells in BALB/c mice showed that palmitic acid could significantly hinder the expansion of transplanted tumors. In this case, palmitic acid also showed the property of enhancing antioxidant activity [21]. The above studies made appropriate explanations and supplements for the cytotoxicity of the essential oil of *T. triquetrum* (L.) Ohashi in this experiment. Moreover, some studies have shown that palmitic acid is a negative regulator of insulin activity and can reduce the degree of insulin-stimulated Akt Ser (473) phosphorylation by inhibiting the activity of SERCA pump ATPase [22]. In more studies, palmitic acid also expressed high anti-MNNQ (anti-N-methyl-N′-nitro-N-nitrosoguanidine) activity in yogurt [23]. In addition, palmitic acid inhibits the activity of thyroglobulin, sodium iodide transporter, and thyroid peroxidase by reducing their expression, which indicates damage to thyroid hormone synthesis [24]. A higher essential oil component of *T. triquetrum* (L.) Ohashi, 1-octen-3-ol, showed broad-spectrum antibacterial activity in its previous volatile compound detection in mushrooms, in which 1-octen-3-ol exhibited effective inhibition of penicillium in soil, and could change the diversity of the soil bacterial community [25]. It is worth noting that 1-octen-3-ol can show strong antioxidant activity in plant extracts [26]. In addition, the antifungal [27], antibacterial [28], and anti-inflammatory activities [29] of 1-octen-3-ol deserve attention. Caryophyllene-rich plants have been widely used in food additives and cosmetics due to their various activities [30]. Caryophyllene, a bicyclic sesquiterpenoid, has significant local anesthetic activity [31,32,33], which makes caryophyllene promising for pharmacological use. In previous studies, caryophyllene promoted the accumulation of drugs through different mechanisms (promoting the intracellular accumulation of paclitaxel-Oregon green, calcein, etc.), which showed a certain degree of anti-cancer activity [34,35]. In addition, the activities of caryophyllene also include antibacterial [36], anti-inflammatory [30], antifungal [37], cytotoxic [38], and antidepressant effects [30]. Although there is no specific study on the activity of (Z)-18-Octadec-9-enolide, there has been a typical experiment for 3-Hexen-1-ol. This one was the most influential component of green aroma [39]. With the increased 3-Hexen-1-ol content, beta-glucosidase treatment can more effectively enhance the fragrance of flowers and herbs and weaken the caramel note [40]. This characteristic has been widely detected and applied. Furthermore, 3-Hexen-1-ol has a noticeable trapping effect on scalable beer species and can be used as an alternative ecological control attractant with a broad-spectrum trapping effect. [41]. Therefore, the application of *T. triquetrum* (L.) Ohashi as a traditional medicine in the treatment of various diseases, as well as in cosmetics, the food industry, etc., is mainly based on the role of the above main ingredients.

*T. triquetrum* (L.) Ohashi belongs to the *T.* Ohashi of Fabaceae. There are only six species of *Tadehagi* in the world, which is relatively simple. Up to now, the research focused to a lesser extent on *T. triquetrum* (L.) Ohashi, which mainly includes alpha-glucosidase inhibitor activity [42], insect-resistant activity [43], and anti-oxidant activity [44].

Determining the antioxidant activity of natural substances mainly depends on chemical and cellular assays. Various measurement methods used to evaluate antioxidant activity depend on different reaction mechanisms. The action mechanism of antioxidants can be generally divided into four types. In laboratory research, scavenging or inhibiting free radicals is more commonly used [45]. Currently, there are many assays available to evaluate antioxidant activity, making it possible to determine antioxidant activity with more than one assay. Taking this into account, it is necessary to objectively evaluate the antioxidant activity of the analyzed substance using reactions based on different mechanisms with various substrates [45,46].

In this study, the essential oils of *T. triquetrum* (L.) Ohashi showed moderate antioxidant activity. Compared with 100 μg/μL standard antioxidants BHT and Trolox, the highest test concentrations of *T. triquetrum* (L.) Ohashi EOs in the ABTS and DPPH experiments had a lower antioxidant effect. The IC_50_ value can reflect the antioxidant activity of *T. triquetrum* (L.) Ohashi. The higher the IC_50_ value, the weaker the ability of *T. triquetrum* (L.) Ohashi to remove DPPH, and the lower its antioxidant activity. The table in the Results Section 2 describes the data obtained by different test methods for EOs. The IC_50_ value for the DPPH assay is 4.36 ± 0.91 mg/mL. According to the analysis of the composition of *T. triquetrum* (L.) Ohashi in this experiment, *T. triquetrum* (L.) Ohashi contains alcohols and ketones to some extent, which can eliminate DPPH· through the HAT mechanism.

The experimental results of DPPH and ABTS showed a positive correlation. The differences between the two methods may be related to the different mechanisms adopted by the two methods (although both are detected by scavenging free radicals) [47]. In the analysis of antioxidant activity through radical scavenging, the HAT (hydrogen atom transfer) mechanism, ET (electron transfer) mechanism, PT (proton transfer) mechanism, and the different tendencies of the above three mixed mechanisms can explain the differences between the DPPH and ABTS results (DPPH prefers HAT mechanism). Based on the single-electron transfer principle [48], the results of the FRAP method significantly differ from those of the other two methods. This is because the DPPH and ABTS methods mainly test substances’ ability to remove free radicals to evaluate the antioxidant activity. In contrast, the FRAP method mainly tests the reduction ability (reducing Fe (III) to Fe (II)) to assess the antioxidant activity [49]. There is a significant positive correlation between alcoholic compounds and the FRAP value [50], which corresponds to the compounds detected in this experiment and antioxidant activity analysis. Zhang et al. note that results obtained by different assays in the experiment were quite different [47], which played a positive complementary role in using more methods to correct the antioxidant activity.

Cholinesterase in the human body mainly hydrolyzes acetylcholine (acetylcholine, as a neurotransmitter, transmits the action potential between two excitable cells) to play the role of weakening the activity of nerve transmission. Anticholinesterase activity can treat myasthenia gravis, sequela of myelitis, and Alzheimer’s disease. Inhibiting the hydrolysis of acetylcholine by acetylcholinesterase (AChE) can effectively prevent the progression of the disease [19]. The antioxidant activity of the *T. triquetrum* essential oils was determined according to the concentration gradient at the highest concentration of 250 μg/mL (the clearance rate was 12.2% at 250 μg/mL). Compared with Huperzine-A as the standard compound, the EOs of *T. triquetrum* (L.) Ohashi showed lower anticholinesterase activity. The active components of drugs for the clinical treatment of Alzheimer’s disease are mainly alkaloids, and the inhibitory activity of other compounds is low [48]. Combined with the identification and analysis of the chemical components of *T. triquetrum* (L.) Ohashi in this experiment, there are almost no alkaloids, so the expression level of anticholinesterase activity is not too high. The anticholinesterase activity also positively correlated with the antioxidant activity. This study complements the application of *T. triquetrum* (L.) Ohashi in pharmacology and makes supplementary explanations for its use in traditional medicine. In the literature survey, no research related to the anticholinesterase activity of *T. triquetrum* (L.) Ohashi was found.

## 4. Materials and Methods

### 4.1. Plant Material

The *T. triquetrum* (L.) Ohashi used in this experiment for substance identification and activity analysis was collected in Qinzhou city, Guangxi province, southern China. As a traditional Chinese herbal medicine, *T. triquetrum* (L.) Ohashi has been partially industrialized and easy to purchase. The correctness of the species was confirmed by comparing the query data with the actual data. Some samples have been preserved at the Ocean College of Shandong University. This experiment mainly used the stalk and leaves of this plant to extract the essential oils. Before the investigation, the *T. triquetrum* (L.) Ohashi had been kept in a cut-and-dried state and stored in a dry place at room temperature.

### 4.2. Extraction of Eseential Oils

During the extraction process, the dried stalk and leaves segments of *T. triquetrum* (L.) Ohashi purchased were weighed (1000 g) and put into a grinder to crush into powder. The *T. triquetrum* (L.) Ohashi powder was put into a round bottom flask with a capacity of 5 L for standby. Then, the appropriate amount of water was added into the round bottom flask and the volatile oil was extracted through water distillation by heating it for about 4 h in the Clevenger-type apparatus. To increase the yield of the extracted essential oil, ether was used to help separate the distilled essential oils from the water layer after extraction (it should be noted that this operation occurred after the experimental device was recovered to room temperature). The essential oil obtained after liquid separation was dried by the termovap sample concentrator and anhydrous sodium sulfate, which can remove the water to obtain a pure organic mixture. The obtained essential oil was put into a glass flask and store it at a low temperature for subsequent testing.

### 4.3. Gas Chromatography–Mass Spectrometry (GC–MS) Analysis

The GC-MS analysis of the essential oils of *T. triquetrum* (L.) Ohashi was carried out on an Agilent 7890–5975 GC-MS instrument (Agilent Technologies, Inc, Santa Clara, CA, USA), and its chromatographic column was HP-5MS-fused silica capillary column (30 m × 250 μmi.d × 0.25 μm film thickness). Under the set gas chromatography conditions (injector temperature: 260 °C; carrier gas: helium, with a flow rate of 1.10 mL/min; heating program setting: the initial temperature was 50 °C kept for 4 min, 6 °C/min to 280 °C and kept for 3 min), and mass spectrometry conditions (EI: 70 eV, 230 °C; mass scanning range: 40–450 Da; acquisition frequency: 2; quadrupole temperature: 150 °C; sample injection volume: 0.5 μL), the relevant detection and analysis were carried out. Under the same conditions, the extracted essential oil and normal paraffin (C7–C30) were analyzed. The Agilent MassHunter Qualitative Analysis 10.0 program was used to confirm the essential oil components, and the normalization of peak area can give us the abundance of compounds we need. Based on the retention time (RT) obtained from the experiment, the corresponding retention index (RI) was calculated according to the retention time. The calculated RI was compared with the index at the peak, which was recorded in the NIST library of the spectral database to complete the determination of the composition of EOs. The composition identification process also referred to the mass spectrum, making the results more accurate and informative.

### 4.4. Antioxidant Activity Test

#### 4.4.1. ABTS Assay

ABTS is a widely used assay to detect the antioxidant activity of scavenging free radicals. The scavenging capability of ABTS^•+^ was evaluated according to the steps in previous studies [49] and modifying some of the steps appropriately. In the experiment, the ABTS^•+^ free radical was produced by mixing 2,2-azinobis-(3-ethylbenzothiazolin-6-sulfonic acid) diammonium salt (ABTS, 7.4 mmol/L) with potassium persulfate (K_2_S_2_O_8_, 2.6 mmol/L). After completing the previous step, the mixture was placed in a dark environment for 12 h to react fully and obtain free radicals after the full reaction. The prepared ABTS^•+^ was diluted with absolute ethanol. In the experimental part of antioxidant activity determination for scavenging free radicals, 150μL ABTS^•+^ solution and 100 μL sample ethanolic gradient diluent (0.1 mg/mL, 0.25 mg/mL, 0.5 mg/mL, 1 mg/mL, 2 mg/mL, and 4 mg/mL) were mixed. The above liquids were mixed in 96-well plates. To improve the reliability of the results, three portions of each concentration for repeated experiments were prepared. After incubation in 96 well-plate for 10 min, the absorbance at 734 nm was read with an Epoch Microplate Spectrophotometer (BioTek, Winooski, VT, USA) and recorded. The inhibition percentage of EOs was calculated with the following formula:$$\mathrm{Inhibition}\%=\left(\frac{A_{0}-A}{A_{0}}\right)\times 100\%$$

#### 4.4.2. DPPH Assay

The DPPH assay is a relatively mature method to evaluate antioxidant activity by measuring free radical scavenging capacity. This test followed the experimental steps of Schlesier et al. [50] and made appropriate corrections to determine the implementable process. The butylated hydroxybenzene (BHT) and 6-hydroxy-2,5,7,8-tetramethylchroman-2-carboxylic acid (Trolox) were used as the positive control. A total of 100 μL of ethanol and 150 μL of prepared 0.17 mmol/L DPPH(2,2-diphenyl-1-picryl-hydrazyl-hydrate) sample solution were added to the microplate as control. Then, 50 μL aliquots of BHT solutions or EOs with different concentrations were added into 200 μL ethanol as blank samples. DPPH was not added in the previous step. The prepared aliquot of the 50 μL BHT solution or the EOs solution was added into the DPPH ethanolic solution to obtain 100 μL mixed solutions, which were placed in 96-well plates. After the above process was completed, the sample to be tested was incubated in the dark for 30 min. After incubation, the absorbance of samples at 516 nm were measured with an Epoch Microplate Spectrophotometer (BioTek), and the readings of each sample at 516 nm were recorded with software. Three parallel experiments were conducted to ensure the accuracy of the experimental results. Finally, the scavenging activity of free radicals (RSA%) was calculated according to the following formula:$$\mathrm{RSA}\left(\%\right)=\left(1-\frac{A_{\mathrm{Sample}}-A_{\mathrm{Sample}\;\mathrm{Blank}}}{A_{\mathrm{Control}}}\right)\times 100\%$$

In the above equation, A_Sample_ represents the absorbance of samples at different concentrations, A_Sample Blank_ represents the absorbance of BHT or EOs ethanolic solution samples without DPPH, and A_Control_ represents the absorbance of the DPPH ethanolic solution. Then, the free radical scavenging activity and the corresponding IC_50_ value can be calculated.

#### 4.4.3. FRAP Assay

In addition to the free radical scavenging test, the antioxidant activity test includes the reduction capacity test. According to the FRAP method described in the literature [16], the steps of reducing chelated ferric iron (Fe (III)) were partially modified to evaluate the antioxidant activity of *T. triquetrum* (L.) Ohashi. The standard solution of Trolox (6-hydrogen-2,5,7,8-tetramethylchroman-2-carboxylic acid) was regarded as the positive control. In addition, Reagent A, acetate buffer solution with a pH value equal to 3.6; Reagent B, 10 mmol/L TPTZ solution; and Reagent C; 20 mmol/L Fe (III) solution, were set. The above working reagents in the proportion of A: B: C = 10:1:1 were mixed to prepare a solution. 1 mol/L and 40 mmol/L were used to acidify the working reagent. A total of 50 μL diluted EOs solution with a concentration of 4000 μg/mL, 2000 μg/mL, 1000 μg/mL, and 500 μg/mL was mixed with a 0.25 mg/mL Trolox solution, whose volumes are 2 μL, 5 μL, 10 μL, 15 μL and 20 μL, respectively. The above solution was thoroughly mixed with 200μL FRAP working reagent in a 96-well plate. Distilled water was used to replace EOs to prepare blank samples through similar steps. The reagents involved in this experiment were tested in three parallel experiments to ensure the reliability of the results, and it was left in the dark for 30 min to react. After 30 min, an Epoch Microplate Spectrophotometer (BioTek) was used to measure the absorbance of the tested sample at 593 nm. The concentration of Fe (II)-TPTZ can be calculated by comparing the absorbance at 593 nm with the Trolox standard curve and the tested solution to evaluate the antioxidant capacity.

### 4.5. Evaluation of Antiacetylcholinesterase Activity

Ellman et al., developed a spectrophotometric method for determining acetylcholinesterase in 1961 [51]. In this experiment, we learned from Ellman’s method of determining anti-acetylcholinesterase activity and made some modifications to make this scheme more feasible. Acetylcholine iodide and butyrylthioline chloride were regarded as reaction substrates in this experiment. 5,5′-dithiobis (2-nitrobenzoic acid) (DNTB) was used to determine the activity of anti-acetylcholinesterase. In the experiment, 140 μL of PBS buffer solution (1 M) with a pH value equal to 8.0, 20 μL of EOs solution, and 20 μL of enzyme solution were added into the standard enzyme plate through a pipette gun and thoroughly mixed. A total of 20 μL PBS buffer solution instead of 20 μL enzyme solution in the above steps was used as control a, and 20 μL 100 μg/mL Huperzine-A solution was prepared with a PBS buffer solution, instead of 20 μL EOs solution, was used as control b in the above steps. In the above steps, the other conditions remained unchanged. At the same time, 160 μL PBS buffer solution and 20 μL enzyme solution were prepared to mix evenly as the blank sample. After the above four groups of reagents were placed on the standard enzyme plate, the samples to be tested had to be incubated at 4 °C for 20 min. After incubation, 10μL DTNB (2 mM) and 10 μL ATCI (15 mM) were added into the samples. The absorbance of samples was measured and recorded at 412 nm with an Epoch Microplate Spectrophotometer (BioTek) after reacting at 37 °C for 20 min. Finally, the inhibition percentage was calculated according to the following formula:$$I\%=\frac{\left(A_{\mathrm{Sample}\;\mathrm{Blank}}-A_{\mathrm{Control}-b}\right)-\left(A_{\mathrm{Sample}}-A_{\mathrm{Control}-a}\right)}{\left(A_{\mathrm{Sample}\;\mathrm{Blank}}-A_{\mathrm{Control}-b}\right)}\times 100\%$$

In the above formula, A_Sample Blank_ represents the activity of the enzyme without the tested sample (phosphate-buffered solution containing PBS with PH = 8), A_Sample_ represents the activity of the enzyme containing the tested sample, and A_Control−a_ represents the activity of the buffer solution. The experiments were performed in triplicate. Huperzine-A used in control b was used as a positive reference.

## 5. Conclusions

In this paper, the chemical compositions of the essential oil from the traditional medicinal plant *T. triquetrum* (L.) Ohashi distributed in southern China were studied. The main chemical constituents of the essential oils of *T. triquetrum* (L.) Ohashi were identified by GC-MS as palmitic acid (22.46%), 1-Octen-3-ol (14.07%), caryophyllene (7.20%), (Z)-18-Octadec-9-enolide (6.04%). According to the analysis, the essential oil of *T. triquetrum* (L.) Ohashi mainly contained alcohol, fatty acids, and olefins. The antioxidant activity of essential oils was dose-dependent. With the increase in the coessential oils content, antioxidant activity also increased. The IC_50_ values determined by the DPPH and ABTS assay were 4.36 ± 0.91 mg/mL and 2.12 ± 0.05 mg/mL, respectively. The antioxidant capacity measured by the FRAP assay was 117.42 ± 8.10 mM/g. The essential oils of *T. triquetrum* (L.) Ohashi showed moderate antioxidant activity when analyzed by the FRAP method, and its antioxidant activity was weakly expressed when determined by the DPPH and ABTS methods. The results supplemented the blank of the antioxidant activity test of the *T. triquetrum* (L.) Ohashi essential oil, and had reference significance. In the study of anti-cholinesterase, EOs expressed a moderated anti-AChE activity, which could weaken the effect of cholinesterase to a certain extent and promote the transmission of neurotransmitters.

In the future, the results of this experiment may be used to conduct relevant biological activity research on other subordinate species of *T.* Ohashi or conduct a deeper evolutionary correlation exploration. The antioxidant and anticholinesterase activities presented in this paper can improve the mechanism of *T. triquetrum* (L.) Ohashi as a traditional medicinal plant. If we want to apply the two types of activity to the human body in biomedicine, we need to supplement more in vivo experimental data for future research. At the same time, the biological activity studied in this experiment can also show a good application prospect in the food and cosmetic industry.

## Figures and Tables

**Figure 1 molecules-28-02734-f001:**
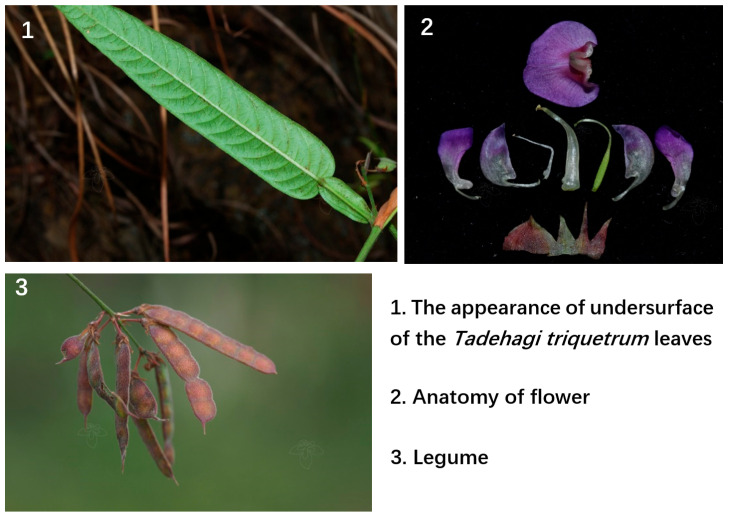
The picture shows the morphological features of the leaves, flowers, and fruits of *T. triquetrum* (L.) Ohashi.

**Figure 2 molecules-28-02734-f002:**
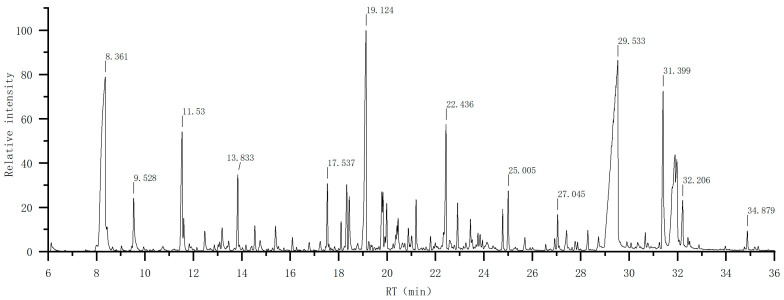
Ion chromatogram of the essential oils of *T. triquetrum* (L.) Ohashi.

**Figure 3 molecules-28-02734-f003:**
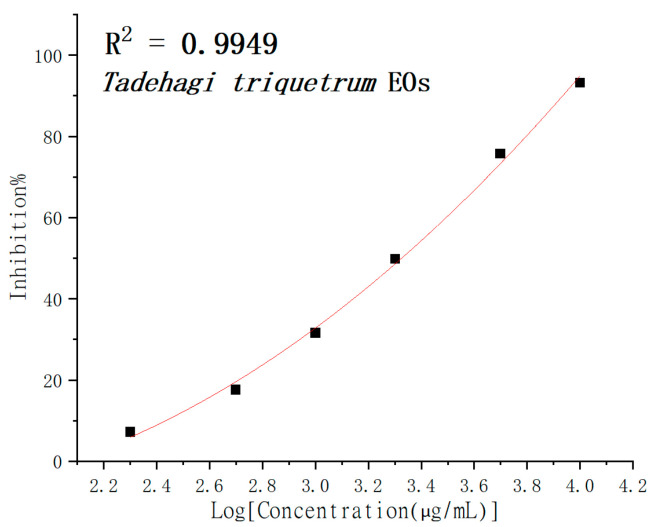
The percentage of the inhibitory effect of the essential oil of *T. triquetrum* in the ABTS assay.

**Figure 4 molecules-28-02734-f004:**
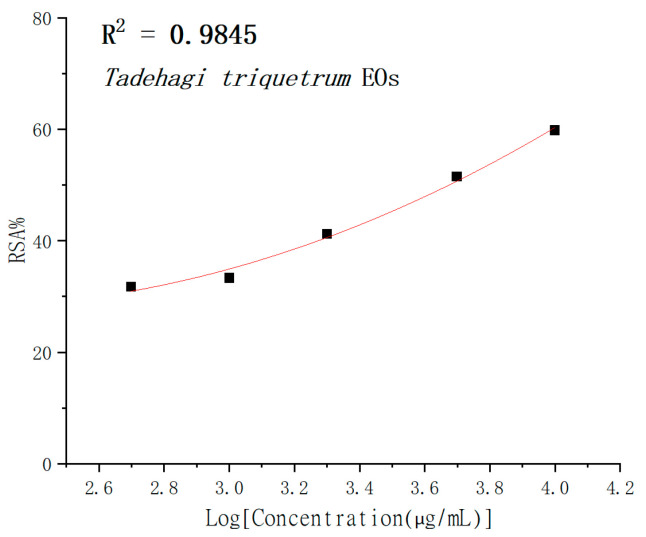
The percentage of free radical scavenging of *T. triquetrum* essential oils in DPPH assay.

**Figure 5 molecules-28-02734-f005:**
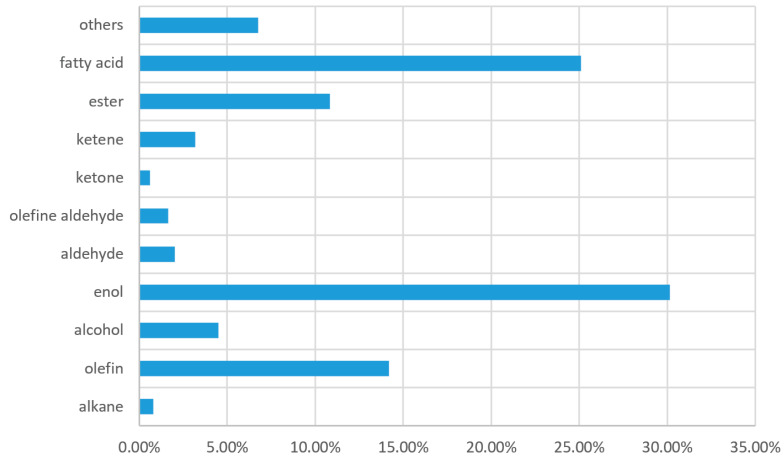
Classification of different compounds in the chemical composition of essential oils from *T. triquetrum* (L.) Ohashi.

**Table 1 molecules-28-02734-t001:** Chemical composition of EOs distilled from *T. triquetrum* (L.) Ohashi.

No.	RT	Compound	RI ^a^	RI ^b^	Area (%)	Identification Method
1	4.602	(E)-2-Hexenal	854	854	0.83%	RRI, MS
2	4.711	3-Hexen-1-ol	858	856	4.60%	RRI, MS
3	5.038	1-Hexanol	871	868	3.53%	RRI, MS
4	6.124	Sorbaldehyde	912	911	0.37%	RRI, MS
5	7.995	3,5,5-Trimethyl-hex-2-ene	978	985	0.23%	RRI, MS
6	8.361	1-Octen-3-ol	989	980	14.22%	RRI, MS
7	8.432	2,2-Dimethylhexanal	991	993	0.74%	RRI, MS
8	9.528	Eucalyptol	1031	1032	1.71%	RRI, MS
9	10.728	(Z)-5-Undecene	1073	1079	0.28%	RRI, MS
10	11.530	Linalool	1101	1099	3.24%	RRI, MS
11	11.601	Nonanal	1104	1104	0.68%	RRI, MS
12	12.463	Pinocarveol	1140	1139	0.47%	RRI, MS
13	13.187	endo-Borneol	1167	1171	0.67%	RRI, MS
14	13.833	α-Terpineol	1191	1189	1.61%	RRI, MS
15	14.536	β-Cyclocitral	1220	1220	0.45%	RRI, MS
16	14.754	Nerol	1230	1228	0.30%	RRI, MS
17	15.392	Geraniol	1257	1255	0.60%	RRI, MS
18	16.086	Bornyl acetate	1285	1285	0.23%	RRI, MS
19	17.537	α-Terpinyl acetate	1350	1350	1.17%	RRI, MS
20	18.099	Copaene	1375	1376	0.53%	RRI, MS
21	18.328	Damascenone	1385	1386	1.36%	RRI, MS
22	18.437	β-Elemene	1390	1391	1.31%	RRI, MS
23	18.786	β-Longipinene	1406	1403	0.29%	RRI, MS
24	19.124	Caryophyllene	1423	1419	7.27%	RRI, MS
25	19.779	Humulene	1455	1454	1.24%	RRI, MS
26	19.828	(E)-β-Famesene	1458	1457	0.77%	RRI, MS
27	19.915	Alloaromadendren	1462	1461	0.28%	RRI, MS
28	19.986	Precocene I	1465	1466	0.76%	RRI, MS
29	20.352	Germacrene D	1482	1481	0.26%	RRI, MS
30	20.401	β-Selinene	1485	1486	0.37%	RRI, MS
31	20.450	β-Ionone	1487	1491	0.60%	RRI, MS
32	20.875	α-Farnesene	1509	1508	0.39%	RRI, MS
33	21.012	γ-Cadinene	1516	1513	0.30%	RRI, MS
34	21.197	δ-Cadinene	1526	1524	0.88%	RRI, MS
35	22.332	Spathulenol	1583	1577	0.37%	RRI, MS
36	22.432	Caryophyllene oxide	1588	1581	2.85%	RRI, MS
37	22.603	Himbaccol	1596	1591	0.29%	RRI, MS
38	22.905	β-Oplopenone	1613	1606	0.82%	RRI, MS
39	23.439	Oxacyclotetradeca-4,11-diyne	1642	1639	0.55%	RRI, MS
40	23.756	α-Cadinol	1659	1653	0.26%	RRI, MS
41	23.838	Precocene II	1663	1658	0.29%	RRI, MS
42	24.121	Aromadendrene oxide-(2)	1678	1678	0.27%	RRI, MS
43	24.776	Pentadecanal	1714	1715	0.60%	RRI, MS
44	25.692	Tetradecanoic acid	1766	1768	0.33%	RRI, MS
45	27.045	Perhydrofarnesyl acetone	1844	1844	0.61%	RRI, MS
46	27.411	Pentadecanoic acid	1866	1867	0.51%	RRI, MS
47	28.289	Farnesyl acetone	1919	1918	0.39%	RRI, MS
48	28.737	Isophytol	1947	1948	0.28%	RRI, MS
49	29.528	Hexadecanoic acid	-	1964	22.70%	MS
50	30.662	Heptadecanoic acid	2069	2071	0.27%	RRI, MS
51	31.399	Phytol	2117	2114	4.22%	RRI, MS
52	31.890	(Z)-18-Octadec-9-enolide	2183	2154	6.10%	RRI, MS
53	31.961	Mandenol	-	2159	3.08%	MS
54	32.206	Octadecanoic acid	2172	2172	1.28%	RRI, MS
55	32.424	Isopropyl oleate	2187	2192	0.24%	RRI, MS
56	34.879	Octadecanamide	2365	2374	0.34%	RRI, MS
57	40.668	Squalene	2831	2827	0.33%	RRI, MS
58	43.739	Hentriacontane	-	3100	0.27%	MS

Concentration calculated from total ion chromatogram; RI ^a^: calculated retention index. RI ^b^: retention index obtained from the mass spectral database. RRI: relative retention indices calculated against n-alkanes; identification method based on the relative retention indices (RRI) of authentic compounds on the HP-5MS column; MS, identified based on computer matching of the mass spectra with Nist/EPA/NIH 2020 mass spectral database and comparison with literature data.

**Table 2 molecules-28-02734-t002:** *T. triquetrum* (L.) Ohashi IC_50_ expression value of antioxidant activity and anticholinesterase activity test.

Activity Test	EOs of *T. triquetrum* (L.) Ohashi	BHT	Trolox
ABTS IC_50_ (mg/mL)	2.12 ± 0.05	0.015 ± 0.0003	0.009 ± 0.0005
DPPH IC_50_ (mg/mL)	4.73 ± 0.91	0.043 ± 0.0012	0.012 ± 0.0008
FRAP Antioxidant Capacity (mM/g)	117.42 ± 8.10		
AChE IC_50_ (mg/mL)	>0.25		

## Data Availability

The data presented in this study are available on request from the corresponding author.

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
