# Peer review of "Chemical Composition and In Vitro Antioxidant Activity and Anti-Acetylcholinesterase Activity of Essential Oils from Tadehagi triquetrum (L.) Ohashi"

_molecules, 2023, doi:10.3390/molecules28062734_

Round 1

Reviewer 1 Report

The article is very interesting and can be published after major revisions

Comments,

Try to make some relationship among the studied methods for antioxidant activities

1.   The antioxidant activity of the essential 15 oils was determined by using ABTS assay, DPPH assay, and FRAP assay, and the chemical compounds identified in the essential oil.

2.  You are dealing with VOC or EO, please clarify.

3.  Why do I see Figure 1 in the introduction section?

4.  I need to see the TIC file of the isolated compounds.

5.  It is not necessary to put the CAS numbers in the table.

6.  Delete figure 3.

7.  Transfer the Figure into the data test and insert them in the table of the identification compounds.  

Reviewer 2 Report

The text is in need of intensive editing. The authors have a poor command of the English, which leads to multiple dramatic errors in the meaning of the content and makes it difficult to understand the content of the manuscript. I have tried to correct some of the sentences to make the meaning of the article clearer. My corrections are listed below. Note the following designations:  <...> - for inclusion and   ]hgdvgv[ - for deletion. The authors need to contact a professional translator.

Abstract:

9: If Tadehagi triquetrum (L.) Ohashi is a plant, it should be added before the name of this organism.

12: The essential oils (EOs) <were>

19: What does mean "the VOCs"? The abbreviation should be deciphered.

Introduction:

33:  ???Are you sure that "natural antioxidants are harmless to the human body"? [3]

40-41: This sentence "The clinical significance of acetylcholinesterase is the main content of liver function examination" is not clear. Did the authors mean the following: "The clinical significance of acetylcholinesterase lies in the fact that its content is the main indicator in the study of liver function" ?

42-44:  This sentence was not written correctly: "Therefore, it is of practical medical .... ".

I suggest to correct it as follows: Therefore, the study of the anti-acetylcholinesterase activity of natural plants is of practical medical importance and opens up the possibility of finding safer drugs.

45-47: This sentence was misspelled in meaning. I suggest to correct it as follows: Although ]there is no clear[  <the> etiology of AD <is unclear>, ]increasing[   <reducing high levels of acetylcholine  ]the acetylcholine amount[ using acetyl- cholinesterase inhibitors has become the most effective treatment

53: What is "Tadehagi (Fabaceae)"? Please, in the beginning of this paragraph, add to this name more information if it is genus or family of plants.

58-60: The phrase is not well understood. I've made some corrections. This is right?: T. triquetrum  (L.) Ohashi has been widely used in folk medicine to ]remove heat and toxic substances[   <lower fevers and expel toxic substances>, ....., and ]eliminate accumulation and killing[  <prevent congestion and kill> insects.

I did not understand the following phrases. What is "to promote dampness"? Is it "to separate phlegm or increase perspiration"? What is "to kill dice"? Is it "to kill mice"?

64: effective ingredients ]for anti-allergic[   <to combat allergic> asthma

77: What is "VOCs"? This must be deciphered.

 Results:

117: All abbreviations must be deciphered the first time they are used in the abstract as well as the first time they are used in the text: ABTS, DPPH, and FRAP

127, 138:  The inscriptions under the abscissa axes in Fig. 4 and Fig. 5 should be given without abbreviations: Log [Concentration (mg/mL)]

146-147:  ... in this experiment ]expressed a specific[   <is of particular>  practical significance.

151: ]were[  <was>  evaluated

 Discussion:

159: In this research   ],[   <of>  the essential oils

167-169: ... in essential oils, ]which[   <so>  T. triquetrum (L.) Ohashi showed palmitic acid chemotype characteristics, which ]had[  <has> far-reaching ...

170: ... algae had ]expressed[ selective cytotoxicity ]to[  <against> human ...

185:  ... by reducing their expression, which  ]expresses the character of damaging the synthesis of[  <indicates damage to>   thyroid hormone  ]activity[  <synthesis>.

186-189:  1-octen-3-ol,  ]as[  a higher   ]content component in the[    volatile oil   <component>   of T. triquetrum (L.) Ohashi, showed broad-spectrum antibacterial activity in <its>   ]the[   previous <volatile compound>  detection ]of volatile compounds[  in mushrooms, in which 1-octen-3-ol ]expressed an [   <exhibited>  effective ...

193-195: Caryophyllene,  ]which belongs to[  <a> bicyclic   <sesquiterpenoid>   ]sesquiterpenoids[, ....., which makes caryophyllene   <promising for pharmacological use>     ]have good pharmacological application prospects[.

212-214: Up to now, the research on T. Ohashi <have> mainly <focused>  ]focuses[  on T. triquetrum (L.) H. Ohashi, <and to a lesser extent>   ]while the research[   on T. triquetrum  (L.) Ohashi   ]is less[, <which> mainly <include>  ]including[   .....

220-221:  ]Because there are many evaluation assays for antioxidant activity, more is needed to identify antioxidant activity by more than one assay.[

<Currently, there are many assays available to evaluate antioxidant activity, making it possible to determine antioxidant activity with more than one assay.>

222-223:  This sentence is difficult to understand. I have corrected the sentence to make it more understandable: Taking this into account, it is necessary to objectively evaluate the antioxidant activity of the analyzed substance using reactions based on different mechanisms with various substrates.

229: As I have understood from the section Results, the T. triquetrum (L.) Ohashi VOCs were less effective antioxidants. Therefore, the sentence [227-229] is discouraging. I have corrected the sentence as follows. Is this correct?

...... T. triquetrum (L.) Ohashi VOCs in ABTS and DPPH experiments ]were lower[  <had lower antioxidant effect>.

232:   ... different test methods ]of[    <for>  VOCs.

243-244:   ...  ]are[ significantly ]different[   <differ>

249:  .... Zhang et al.  ]expressed[   <note> ...

266:  .... ]was[ also positively correlated ....

268: use  ]as[  <in> traditional

Conclusions:

406:   ]expressed[  <was> dose-dependent

407:   ]was[   also increased

410:     ]in[  <when analyzed by>  the FRAP method, and its antioxidant <activity> was weakly expressed  ]in[   <when determined by> DPPH and ABTS methods.

419:  the two <types of activity>   ]activities[   <to the human body in biomedicine>  ]to biomedicine  acting on the human body[

Reviewer 4 Report

Manuscript "Chemical Composition and in vitro Antioxidant Activity and Anti-Acetylcholinesterase Activity of Essential Oils from Tadehagi triquetrum (L.) Ohashi" is well-written, interesting and will be valuable for other scientist. It could be accepted for publication. I have only some comments.

1. There is no explanation of the abbreviation "VOCs"

2. I have doubts about taxonomy. Is it really T. triquetrum (L.) Ohashi not the same as T. triquetrum (L.) H. Ohashi? Moreover, "T. Ohashi" (line 211) is misleading, as it seems like a name of species, but it show genus ("T." is for genus Tadehagi) followed authority ("Ohashi" and should not be italicisied). This should be reviewed throughout all manuscript.

3. Line 360: replace "TPTX" with "TPTZ".

4. "in vivo" and "in vitro" should be italic

Round 2

Reviewer 1 Report

Minor revision before publication

Delete Figure 5. Classification of different compounds in the chemical composition of essential oils from 181 T. triquetrum (L.) Ohashi., and put these data in table 1.

Reviewer 2 Report

There left a few errors in the manuscript. See my corrections listed below. Note the following designations:  <...> - for inclusion and   ]hgdvgv[ - for deletion.

 Abstract:

9: The Journal is intended for a wide audience, not just narrow specialists. Therefore, to make the content accessible to readers, the authors should add the word "plant" before the name Tadehagi triquetrum (L.) Ohashi.

12: In the abstract, since essential oils (EOs) have been deciphered once, the abbreviation "EOs", without "essential oils", should be used hereafter.

Introduction:

57: which <is> represented

76: in  ]pathological[ research of   <pathologies>

Results:

122: Figure-3 caption has been removed. The Figure-3 caption must be added.

141, 154:  The inscriptions under the abscissa axes on the left panels in Fig. 3 and Fig. 4 should be given without abbreviations: Log [Concentration (mg/mL)], as on the right panels.

 Discussion:

230-232:  Some parts of the sentence were lost. Authors must improve the sentence.

Reviewer 3 Report

Only one for R397: Change “Leave in the dark” in “They were left in the dark”.